Association between iscR-based phylogeny, serovars and potential virulence markers of Haemophilus parasuis

Li Junxing
Xu Lihua
Su Fei
Yu Bin
Yuan Xiufang yuanxf@zaas.ac.cn
Zhejiang Academy of Agricultural Sciences, Institute of Animal Husbandry and Veterinary Medicine , Hangzhou, Zhejiang , China
Tulkens Paul
Electronic publication date: 2019 May 14
Publication date: 2019
Volume: 7
Electronic Location ID: e6950
Received 2019 Jan 4; Accepted 2019 Apr 9
Copyright: © 2019 Li et al.
Copyright year: 2019
Copyright holder: Li et al.
License: This is an open access article distributed under the terms of the Creative Commons Attribution License, which permits unrestricted use, distribution, reproduction and adaptation in any medium and for any purpose provided that it is properly attributed. For attribution, the original author(s), title, publication source (PeerJ) and either DOI or URL of the article must be cited.
License URL: https://creativecommons.org/licenses/by/4.0/

Keywords: Serovar, iscR, Population structure, Haemophilus parasuis, Virulence

Funding: National Natural Science Foundation of China 31201934 Zhejiang Provincial Key Research and Development programs 2019C02052-2 Zhejiang Provincial Science and Technology Plans 2016C02054-7 Zhejiang Academy of Agricultural Sciences 10102000617CF0101G/003/003 This work was financially supported by the National Natural Science Foundation of China (31201934), the Zhejiang Provincial Key Research and Development programs (2019C02052-2), the Projects of Zhejiang Provincial Science and Technology Plans (2016C02054-7) and the Young Talent Training Project of Zhejiang Academy of Agricultural Sciences (10102000617CF0101G/003/003). There was no additional external funding received for this study. The funders had no role in study design, data collection and analysis, decision to publish, or preparation of the manuscript.

==============================
Haemophilus parasuis is an economically important bacterial pathogen of swine. Extensive genetic and phenotypic heterogeneity among H. parasuis strains have been observed, which hinders the deciphering of the population structure and its association with clinical virulence. In this study, two highly divergent clades were defined according to iron–sulphur cluster regulator (iscR)-based phylogeny analysis of 148 isolates. Clear separation of serovars and potential virulence markers (PVMs) were observed between the two clades, which are indicative of independent evolution of the two lineages. Previously suggested virulence factors showed no correlation with clinical virulence, and were probably clade or serovar specific genes emerged during different stage of evolution. PVMs profiles varied widely among isolates in the same serovar. Higher strain diversity in respect of PVMs was found for isolates from multi-strain infected farms than those from single strain infected ones, which indicates that multi-strain infection in one farm may increase the frequency of gene transfer in H. parasuis. Systemic isolates were more frequently found in serovar 13 and serovar 12, while no correlation between clinical virulence and iscR-based phylogeny was observed. It shows that iscR is a reliable marker for studying population structure of H. parasuis, while other factors should be included to avoid the interference of gene exchange of iscR between isolates. The two lineages of H. parasuis may have undergone independent evolution, but show no difference in clinical virulence. Wide distribution of systemic isolates across the entire population poses new challenge for development of vaccine with better cross-protection. Our study provides new information for better deciphering the population structure of H. parasuis, which helps understanding the extreme diversity within this pathogenic bacterium.

Introduction

Haemophilus parasuis is a member of the family Pasteurellaceae, and is the causative agent of Glässer’s disease in swine. H. parasuis could be a commensal organism of the upper respiratory tract of healthy pigs or pathogens causing systemic infection (Biberstein, Gunnarsson & Hurvell, 1977). Extensive heterogeneity among H. parasuis strains have been observed, and increasing efforts are made in deciphering the population structure and its possible relevance with clinical outcome (Oliveira, Blackall & Pijoan, 2003; Howell et al., 2014; Turni, Singh & Blackall, 2018; Zhao et al., 2018).

Strain classification of H. parasuis has been widely studied serotypically, and genotypically, since differentiation of non-virulent strains from virulent strains is essential for diagnosis and control of the disease (Kielstein & Rapp-Gabrielson, 1992; Turni & Blackall, 2005; Olvera, Calsamiglia & Aragon, 2006a; Moreno et al., 2016).

Serotyping is the most commonly used subtyping method, and is traditionally considered to be associated with virulence, but increasing evidence indicated that serovar is a poor proxy for virulence (Aragon et al., 2010; Brockmeier et al., 2014; Lawrence et al., 2014). Sequencing studies of hsp60 and 16s rRNA gene revealed two clades of strains in H. parasuis and the presence of a separate lineage of virulent strains (Olvera, Calsamiglia & Aragon, 2006a; Angen et al., 2007). Population structure analysis at genome level revealed the separation of H. parasuis into two clades, which were further divided into five Bayesian analysis of population structure sub-clade, but no influence of disease association on population structure was observed (Howell et al., 2014). Hence, strains in certain sub-type tend to be virulent, but the relationship between sub-type and virulence phenotype is still not clear.

An increasing number of potential virulence markers (PVMs) have been identified for H. parasuis, of which some were already used for development of molecular methods for virulence prediction (Sack & Baltes, 2009; Zhou et al., 2010; Wang et al., 2011; Yu et al., 2014; Galofre-Mila et al., 2017; Howell et al., 2017). As clinical data on these PVMs are still limited, its clinical relevance needs to be further confirmed.

Iron–sulfur clusters are essential protein cofactors that participate in numerous biological processes in most organisms (Santos, Pereira & Macedo-Ribeiro, 2015). Iron–sulphur cluster regulator (iscR), a member of the Rrf2 family transcriptional regulators, is a global transcription regulator involving in various physiological processes during bacteria growth and stress responses. Growing evidence shows that iscR governs the proper regulation of virulence factors in many bacterial pathogens such as type III secretion system of Yersinia pseudotuberculosis, capsular polysaccharide biosynthesis of Klebsiella pneumonia, antigen I fimbriae of enterotoxigenic Escherichia coli and catalase A activity of Pseudomonas aeruginosa (Haines et al., 2015; Miller & Auerbuch, 2015; Santos, Pereira & Macedo-Ribeiro, 2015). IscR deficient mutant displays dramatic attenuation of virulence in several pathogens, and the important role of iscR in bacterial pathogenesis is becoming increasingly apparent (Crack et al., 2012; Miller & Auerbuch, 2015).

The iscR gene was found in reference strains of all 15 serovars, but little is known about the role of iscR in pathogenicity of H. parasuis (Yue et al., 2009; Howell et al., 2013). IscR is a highly conserved protein among Proteobacteria and particularly within a certain species, but apparent amino acid substitutions were observed among 15 reference strains of H. parasuis (Howell et al., 2013; Mettert, Perna & Kiley, 2014; Santos, Pereira & Macedo-Ribeiro, 2015). As iscR is a global regulator of virulence factors in many pathogenic bacteria, it is essential to evaluate the possible impact of the amino acid substitutions in iscR protein on population structure and clinical virulence of H. parasuis. In this study, population structure of H. parasuis was analyzed based on the phylogeny, serovars and PVMs profiles of 148 isolates. Meanwhile, potential correlation among population structure and clinical phenotype was explored.

Materials and Methods

Isolates and culture conditions

A total of 148 H. parasuis strains were used in this study, of which 145 strains were isolated from clinical samples sent to our diagnostic center from September 2009 to January 2017, and the other three strains were isolated from nasal cavity (Dataset S1). The isolates were originated from 125 independent pig farms in three provinces of eastern China. A sterile loop was used to inoculate TSA plates (supplemented with 2.5% bovine serum and 0.05% NAD) from samples of nasal cavity, lung, spleen, joint, brain, pericardium liquid, pleural effusion, and seroperitoneum. The plates were kept in a 37 °C incubator under humidified 5% CO2 for 24–72 h until the colony appears. Unequal number of colonies (from four to seven) was picked based on the number of samples, isolation sites or morphology of the colony for each independent farm, and each selected colony was inoculated on another TSA plate for further purification. Then, the isolates were transferred from TSA plates to TSB broth (supplemented with 2.5% bovine serum and 0.05% NAD) for bacteria enrichment.

Serovar determination

Genome DNA of H. parasuis was extracted by digestion with proteinase K, and the isolates were identified by a species specific PCR test (Angen et al., 2007). Serovars of the isolates were determined by the two newly developed molecular serotyping methods, which were faster, more sensitive, and more specific than the traditional IHA (Howell et al., 2015; Jia et al., 2017). Using of the two molecular serotyping methods in combination allows the differentiation of all the 15 serovars. If all the picked colonies from one independent farm were determined to be the same serovar, then one of colonies was selected to be the representative. If more than one serovar were detected in a farm, then one representative for each different serovar was selected for further analysis. The designation of each clinical isolate includes information of province (ZJ for Zhejiang province), year (09 for year of 2009), sequence number (01 for the first isolates of the year) of isolation, and serovar (S1 for serovar 1 etc., Su for unknown serovar) of the isolate.

IscR sequencing and analysis

Primers for amplification of complete open reading frame of iscR gene were designed based on the iscR gene sequence of the strain SH0165 (GenBank accession number: CP001321) (Xu et al., 2011), and listed in Table 1. Amplicon of each isolate was purified and cloned into pMD 19 T vector, and send to TSINGKE Biological Technology Co., Ltd for gene sequencing. IscR gene for each isolate was sequenced twice in both the forward and reverse direction. IscR sequences of 15 reference strains (GenBank accession numbers: KC795297, KC795318, KC795335, KC795364, KC795350, KC795383, KC795404, KC795422, KC795440, KC795459, KC795480, KC795495, KC795512, KC795532, KC795549) of H. parasuis were originated from a previous study (Howell et al., 2013). Alignment of the iscR sequence was prepared with MEGA 7.0, Lasergene 7 or DNAMAN 9. Phylogenetic tree was decorated with an online tool iTOL v4.2.3 (https://itol.embl.de/).

Table 1 Primers used for the PCR assays for potential virulence markers and iscR.

Gene	Primer name	Primers(5′-3′)	Amplicon size (bp)	References	
HAPS_0254	B14-F
B14-R	ACACCTTATGCTTCCGCTAT
ACGGTAACAGAACAAGAGCC	146	Wang et al. (2011)	
HAPS_0254	D45-F
D45-R	CTTCCGCTATCGCATTAA
AACAGAACAAGAGCCTAAA	131		
nhaC	E30-F
E30-R	GTCCAGGAAGCATAATACA
TACAAGGTGGCGAGATAA	312		
fhuA	E35-F
E35-R	TCTAAGCGATGGGATTGAGC
GGTGGCGTAAGACGTGATT	461		
hhdA	MP_A1
MP_A2	GGTTCTAGTTCACAAACAGCCAATAC
GATATTTACCCCTGCCTTCATTGTATC	964	Sack & Baltes (2009)	
hhdB	MP_B1
MP_B2	ATCTTGCCCTGATTAGAGAGTAGGAGT
GTGAATATAGCCCTTATCCAAATAGGC	557		
hsdR	hsdR-F
hsdR-R	ATGACTATTGAAACAACGCCTATCA
GAACTTGGTTAAAGGCTTCC	544	Yu et al. (2014)	
hsdS*	hsdS-F
hsdS-R	ATGATAGAAAGTCGTTTTATTGAAAAG
GAAAACTTAATAATTTCTCTCTGTA	1,219	This study	
ompP2	ompP2-F
ompP2-R	ATGAAAAAAACACTAGTAGCA
TTACCATAATACACGTAAACC	1,092		
HPS_21058	21058-F
21058-R	CCGAAAGCATAGATCCAAATGC
CCACCTTGTTTACTTGCTTCTGC	590	Howell et al. (2017)	
HPS_21059	21059-F
21059-R	CGTAGCATACGCACACCTAAAG
GAAAGGGCAATAGATACATTTCGG	720		
vtaA	AV1-F
V1-R
NV1-R	AAATATTTAGAGTTATTTGGAGTC
AATATACCTAGTAATACTAGACTTAAAAG
CAGAATAAGCAAAATCAGC	190
222	Galofre-Mila et al. (2017)	
iscR	iscR-F
iscR-R	ATGAAATTAACTTCACGAGGACG
TTAATGGTGATGATCGTGGCAG	447	This study	
Note:

* Primers of the gene were newly designed in this study and were different from the original one.

Clinical backgrounds and detection of PVMs

Isolates from systemic sites such as spleen, brain, pericardium liquid, pleural effusion, seroperitoneum, and joint were considered to be systemic ones, and those from lungs and nasal cavities were considered to be respiratory ones and nasal ones, respectively. To further confirm the population structure and its relevance with clinical phenotype, 12 PVMs from five pieces of literatures were selected. Primers of one of the selected PVMs (hsdS) were newly designed in this study (Table 1), and primers of other selected PVMs are the same with that in the literatures (Sack & Baltes, 2009; Wang et al., 2011; Yu et al., 2014; Howell et al., 2017; Galofre-Mila et al., 2017). The presence or absence of the selected PVMs in all the isolates were determined by PCR. Two or more strains from the same farm with any difference in respect of serovar, iscR clade, presence or absence of any PVMs were considered to be different isolates. The differences in average number of PVMs between serovars were determined by One-way ANOVA with SPSS Statistics 17.0. Differences were considered statistically significant when P < 0.05.

Accession numbers

The nucleotide sequences of iscR for the 148 isolates were submitted to GenBank. The accession numbers for the individual CDSs are MH982282–MH982429.

Results

Separation of clinical isolates into two deeply divergent groups by iscR-based phylogenetic analysis

Iron-sulfur cluster regulator gene was presented in all 148 isolates used in this study with the consistent open reading frame size of 447 bp. Phylogenetic sequence analysis of iscR amino acid sequences showed that the isolates were divided into two deeply divergent clades, which were designated as clade I and clade II (Fig. 1). Dominant sequence type (ST1 and ST6) in each clade were further aligned with iscR of other Gram-negative bacteria, and simultaneous amino acid substitutions at highly conserved residue 29, 68, 77 made clade I H. parasuis an evolutionarily distant lineage from this and other Gram-negative species (Fig. 2A). Amino acid sequence comparison revealed nine sequence types from the clinical isolates, and amino acid substitutions at residue 29, 68, 70–71, 77–78, 140 took place at 100% or nearly 100% between iscR of the two clades (Fig. 2B). Four iscR protein sequence types were identified from the 15 reference strains, of which three were identical to those from the clinical isolates in this study (Fig. S1).

Figure 1 Phylogenetic tree of iscR proteins from 148 isolates.

The proteins were highly conserved within both of the clade but deeply divergent between the two clades. Molecular Phylogeny was analyzed by Maximum Likelihood method using MEGA7 and decorated with an online tool iTOL v4.2.3 (https://itol.embl.de/).

Figure 2 Alignment of iscR protein sequences of H. parasuis.

(A) Conservation of iscR proteins from Gram-negative bacteria. IscR protein of the dominant sequence type in each clade (ST1 for clade I and ST6 for clade II) was aligned. Residue numbering refers to the H. parasuis iscR sequence. DNA-binding domain, encompassing the HTH motif and a wing, was indicated with arrows. Fe/S cluster ligands were indicated with triangles. Strictly conserved amino acids were highlighted in dark blue, and decreasing residue conservation was represented by a color gradient from red to green. Alignment was prepared with DNAMAN 9. (B) Representatives of the nine sequence types were aligned using Lasergene (MegAlign). Amino acid residue of the sequence type 1 (ST1) were used as marker, and residues that match ST1 exactly were hide (as “.”). The number of isolates represented by each sequence type is shown in the brackets. The clades distribution of each sequence type were indicated.

Of all the isolates used in this study, clade I isolates were more prevalent (62.2%) than clade II isolates (37.8%). A higher proportion of isolates were systemic ones in clade I (44.6%) than that in clade II (30.4%) (Fig. 3A). No tissue tropism was seen in respect of iscR-based phylogeny, as isolates from all the systemic sites can be found in both clades (Fig. 3B). It is noteworthy that 93.8% of the isolates from pericardial effusion are belong to clade I (Fig. 3B).

Figure 3 Clade distribution (A) and isolation site (B) of systemic isolates.

The number of isolates are indicated with data tags in the histogram. (A) Distribution of nasal, respiratory, and systemic isolates in clade I and clade II. (B) Isolation site of systemic isolates in clade I and clade II.

Clear correlation between serovar and iscR-based phylogeny

The two molecular serotyping methods allow the assignment of 144 (97.3%) isolates to 13 serovars except serovar 3 and 15, and 4 isolates (2.7%) could not be assigned to any of the 15 serovar known so far. Serovar 4 (18.9%) and serovar 12 (18.9%) were the most prevalent followed by serovar 13 (16.2%) and serovar 5 (13.5%) (Fig. 4A). Isolates within most of the serovars tend to be distributed in the same clade. Serovars 4, 5, and 14 were dominated by clade I isolates, while serovars 1, 2 and 7 were dominated by clade II isolates (Fig. 4A). Although slightly higher proportion of serovar 12 and 13 isolates were found in clade I, the distribution of isolates in the two serovars was relatively equal in both clades. For the remainder of the serovars, the number of isolates was too low to be able to restrict them to one clade.

Figure 4 IscR-based phylogenic (A) and clinical relevance (B) of isolates in each serovar.

(A) Clade distribution of isolates in each serovar. (B) Number of systemic, respiratory, and nasal isolates in each serovar.

Systemic isolates were found in 10 of the 13 serovars identified. Of all the serovars with not less than five isolates, systemic isolates were more common in serovar 13 (54.2%) and serovar 12 (50%), and similar proportion of systemic isolates were found in serovar 1, 4, 5 ranging from 35% to 37.5%, which were followed by serovar 7 (20%) and serovar 14 (14.3%) (Fig. 4B). These data indicate that virulence varies among serovars, and serovars 13 and 12 are more strongly linked to systemic infections.

Presentation of PVMs showed obvious correlation with iscR-based phylogeny and serovar

To further confirm the population structure and its relevance with clinical phenotype, distributions of 12 previously identified PVMs were investigated in all the 148 isolates. OmpP2 and vtaA genes could be detected in 100% and 98.6% of the isolates, respectively. Only four isolates were identified to be avirulent by the vtaA method, of which one was isolated from joint of diseased pig. Hence, presence of ompP2 and vtaA gene showed low resolution of the isolates, and the other 10 PVMs were selected in the following studies.

The positive rates of most of the PVMs were similar among serovars (4, 5, 12, 13, and 14) which were dominated by isolates in clade I and among serovars (1, 2, and 7) which were dominated by isolates in clade II. The positive rates were clearly different when compared between serovars of one group (serovars 4, 5, 12, 13, and 14) and the other group (serovars 1, 2, and 7). Most of the PVMs (8/10) were more prevalent in clade I than in clade II (Fig. 5). Of all the serovars with not less than 5 isolates, serovar 5 isolates carried the most PVMs with the average number of 7.35, while serovar 1 isolates carried the least with an average number of 1.5. Serovars dominated by clade I isolates were found with presence of remarkable higher average number of PVMs than those dominated by clade II isolates (P < 0.05) (Fig. 6). Interestingly, average number of PVMs in serovar 12 and 13 were similar to the serovars dominated by clade I isolates (Fig. 6), although the distribution of isolates of the two serovars was relatively equal in both clades. In this study, only one isolate was found with the presence of all the 10 selected PVMs (Fig. 7).

Figure 5 Distribution of potential virulence markers in each serovar.

Percent of isolates with the presence of potential virulence markers in each serovar.

Figure 6 Average number of potential virulence markers (PVMs) carried by the isolates in each serovars.

Data were shown as mean + S.D. A P-value was computed using the one-way ANOVA with SPSS Statistics 17.0. Variations were considered statistically significant when the P-value was ≤0.05, which is indicated with “*”.

Figure 7 Maximum likelihood phylogenetic tree with the 148 isolates of H. parasuis, based on iscR protein sequence alignment.

The tree was constructed with MEGA7, and decorated using ItoL (https://itol.embl.de/). Clade colors indicate the divergence of clade I (red) and clade II (blue). Inner circle represents lineage for each strains, and background color of the text label at each node indicates the serovar of the strain. The middle circle shows the isolation site of the strain. The outer circle shows the presence (red) or absence (green) of the 10 selected potential virulence markers (PVMs) in each isolates.

Higher strain diversity in isolates from multi-strain infected farms

A total of 34 PVMs profiles were identified based on the presence or absence of the 10 selected genes (Table 2). The most prevalent profile was shared by 39 isolates, followed by 22 isolates in profile 20. More than half of the profiles (58.8%) were represented by only one isolate, which suggests high gene heterogeneity in accessory genome among isolates. Of the six profiles with no less than five isolates, systemic isolates were most prevalent in profile 27 (4/5) and least prevalent in profile 29 (0/10). Similar proportion of systemic isolates (29–46%) were observed in other profiles (Table 2). Moreover, 10.2% of the systemic isolates do not carry any of the 10 selected PVMs (Table 2), and all of which were distributed in clade II with the exception of one serovar unknown isolate (Fig. 7). Neither presence of any of the selected PVMs nor the number of PVMs carried by the isolates showed correlation with clinical phenotype of systemic infection (Fig. 7).

Table 2 Potential virulence markers (PVMs) profiles and systemic isolates distribution in each profile.

No. of PVMs	Serial number of the profiles	No. of systemic/total isolates in the profile	HAPS_0254 (B14)	HAPS_0254 (D45)	nhaC	fhuA	hhdA	hhdB	hsdS	hsdR	HPS-21058	HPS-21059	
0	1	6/13	−	−	−	−	−	−	−	−	−	−	
1	2	2/7	−	−	−	−	−	−	−	−	+	−	
3	1/1	−	+	−	−	−	−	−	−	−	−	
2	4	0/1	−	−	−	−	−	−	−	+	+	−	
5	2/3	−	−	−	−	−	−	−	−	+	+	
6	0/1	−	−	−	−	−	−	+	+	−	−	
7	0/1	−	−	+	−	+	−	−	−	−	−	
3	8	0/1	−	+	−	−	+	−	−	−	+	−	
9	0/1	−	−	−	−	−	−	−	+	+	+	
10	0/1	+	+	−	−	−	−	−	+	−	−	
11	0/1	+	+	−	−	+	−	−	−	−	−	
4	12	0/1	−	−	+	−	+	+	−	+	−	−	
13	2/2	+	+	+	−	−	+	−	−	−	−	
14	1/1	−	+	−	−	+	−	−	+	+	−	
5	15	0/1	−	+	+	−	+	+	−	−	+	−	
16	1/1	+	+	+	−	+	−	−	+	−	−	
17	1/1	+	+	+	−	+	+	−	−	−	−	
18	2/2	+	+	+	−	−	+	−	+	−	−	
19	1/3	−	−	+	−	+	+	+	+	−	−	
6	20	9/22	+	+	+	+	+	+	−	−	−	−	
21	1/1	−	−	+	+	+	+	+	+	−	−	
22	3/4	+	+	+	−	+	+	−	+	−	−	
23	1/1	+	+	+	+	−	+	−	+	−	−	
24	1/1	−	+	+	−	+	+	−	−	+	+	
25	0/1	+	+	+	−	+	−	+	+	−	−	
26	0/1	+	+	−	−	+	+	+	+	−	−	
7	27	4/5	+	+	+	+	+	+	−	+	−	−	
28	13/39	+	+	+	−	+	+	+	+	−	−	
29	0/10	+	+	+	+	+	+	−	−	−	+	
8	30	1/1	+	+	+	+	+	+	−	+	+	−	
31	5/13	+	+	+	+	+	+	+	+	−	−	
9	32	1/3	+	+	+	+	+	+	+	+	+	−	
33	0/2	+	+	+	+	+	+	+	+	−	+	
10	34	1/1	+	+	+	+	+	+	+	+	+	+	
Total	34	59/148		

Of all the 125 independent pig farms, multi-strain (two or three) infection was found in 16% of the pig farms and single strain infection was found in the other 84% pig farms. An average of every 4.4 isolates from the single strain infection farm produce one PVMs profile, while every 2.2 isolates from the multi-strain infection farm produce one PVMs profile (Table 3). Hence, multi-strain infection in one farm resulted in more diverse PVMs profiles in those isolates, which indicates that frequent gene transfer among isolates took place in multi-strain infected farms.

Table 3 Potential virulence markers (PVMs) profiles in single and multi-strain infected farms.

Infection mode	No. of farms (%)	No. of isolates	No. of PVMs profiles	
Single strain	105 (84)	105	24	
Multi-strain	20 (16)	43	20	

Potential gene exchange at the capsule locus

To study the possible recombination between strains at the capsule locus, all isolates were clustered based on pattern similarity of PVMs. Two distinct clusters were identified, and were designated as cluster A and cluster B (Fig. 8). Most of the isolates (92.1%) of serovar 4, 5, 12, 13, and 14 harbored more than five PVMs, and were distributed in cluster B. Most of the isolates (78.3%) of serovar 1, 2, and 7 only have less than two PVMs, and were listed in cluster A (Fig. 8). Coincidently, clear separation of isolates from those serovars (4, 5, 12, 13, and 14) and other serovars (1, 2, and 7) was observed in iscR-based phylogeny analysis (Fig. 4A). It indicates that these two clusters were possibly originated from two independently evolutionary lineages, which may have branched off from each other for a long evolutionary period.

Figure 8 Hierarchical clustering of H. parasuis isolates based on the pattern similarity of potential virulence factors.

Isolates were ordered along the columns based on the similarity in the presence (red) and absence (black) pattern of 10 potential virulence factors (HAPS_0254 (B14), HAPS_0254 (D45), nhaC, fhuA, hhdA, hhdB, hsdS, hsdR, HPS_21058, and HPS_21059). The similarity matrix was calculated using the Euclidean distance. Cluster analysis was performed by the Hierarchical clustering (HCL) through the cluster analysis procedure of MeV (Multiple Experiment Viewer) v4.9.0 package.

Recombination event may take place when two or more strains co-exist in one farm, especially when in the same infected organ or commensal site. A small portion of isolates (21.7%) of serovar 1, 2, and 7 such as ZJ1510-S1 and ZJ1310-S7 were divided into cluster B. Similarly, a small portion of strains (7.9%) of serovar 4, 5, 12, 13, 14 were listed in cluster A, for instance, ZJ1604-S4, ZJ1207-S13, ZJ1405-S13, and ZJ1424-S13 etc., (Fig.8). All these isolates harbored PVMs characteristic that do not match the majority of isolates of the same serovar, but consistent with PVMs characteristic of serovars which were mainly distributed in the other cluster. It indicates that these isolates may have undergone gene exchange between one serovar group (1, 2 or 7) and the other serovar group (4, 5, 12, 13, and 14) at or around the site of the capsule locus. Hence, isolates in the same serovar can vary greatly in terms of PVMs, which could lead to differences in the pathogenicity and immunogenicity within one serovar. It would be more difficult to identify in this study when exchange of capsule locus take place between serovars with similar PVMs characteristics.

Discussion

Phylogeny and population structure of H. parasuis were widely studied, and H. parasuis strains could be divided into two groups in a variety of ways (Oliveira & Pijoan, 2004; Olvera, Cerda-Cuellar & Aragon, 2006b; Turni & Blackall, 2010; Howell et al., 2014; Moreno et al., 2016; Wang et al., 2016). Results of current study on correlation between population structure and virulence were contradictory. H. parasuis strains in some sub-groups tend to be virulent (Ruiz et al., 2001; Olvera, Calsamiglia & Aragon, 2006a; Olvera, Cerda-Cuellar & Aragon, 2006b; Wang et al., 2016), while no association between population structure and clinical virulence was found in others (Zehr, Lavrov & Tabatabai, 2012; Boerlin et al., 2013; Howell et al., 2014). No strong correlation between clinical phenotype and iscR-based phylogeny or PVMs profiles was observed in this study.

The relationship between serotype and other typing methods has been evaluated (Turni & Blackall, 2010; Ma et al., 2016; Zhao et al., 2018), but correlation were only been identified in a few reports (Boerlin et al., 2013; Howell et al., 2014). In this study, isolates of serovar 4, 5, and 14 were found predominantly distributed in clade I, and those of serovar 12 and 13 were mostly found in clade I, while those of serovar 1, 2, and 7 were predominantly in clade II. It is similar to the results that separation of serovars (serovar 5, 12, 13 predominantly in clade I and serovar 7 in clade II) was found between the two clades in a previous study (Howell et al., 2014). Our study was also consistent with another report, in which serovar 5, 13, 14 were found only in cluster I and serovar 2 was only found in cluster II (Boerlin et al., 2013).

Systemic isolates were most frequently seen in serovar 13 in this study. In a previous study, serovar 13 were not the most prevalent one in respect of total isolation frequency, but was the most prevalent one with regard to the frequency of systemic isolates (Cai et al., 2005). Serovar 5 and 4 were determined to be more strongly linked to disease in another study (Howell et al., 2014). Mega data on correlation between serovar and virulence or clinical phenotype is still too limited to understand the relationship. Moreover, virulence of isolates varies greatly within the same serovar (Lawrence et al., 2014; Yu et al., 2014), and the reason for the variation is still to be determined. In this study, some isolates carry completely different PVMs compared to most of the counterparts within the same serovar, which indicates that those isolates may vary extensively in the accessory genome. The difference in the accessory genome may be responsible to the variation of virulence within one serovar. Genes in the accessory genome had been linked to differences in clinical phenotype (Howell et al., 2014). The vast genetic heterogeneity among isolates may pose a challenge to vaccine development (Turni, Singh & Blackall, 2018).

In this study, selected PVMs were similarly distributed in serovars of which the isolates were predominantly distributed in the same clade, but huge differences were found when compared in between the clades. Hence, similarity of PVMs distribution in the accessory genome further suggest the close relationship among serovars dominated by isolates in the same clade. Moreover, phylogenetic analysis of gene content of the capsule loci support the close relationship among serovar 1, 2, and 7 (Howell et al., 2013), which is consistent with our finding that isolates of those serovars were all found in the same clade with exception of one isolate of sevovar 2. The coincidence of iscR-based phylogeny, separation of serovars and similarity of PVMs distribution in the accessory genome strongly suggests the independent evolution of the two branches of H. parasuis. The phylogenetic and amino acid substitution analysis of iscR inferred that clade II was the first to branch off from its ancestor, and clade I was possibly originated from clade II at the early stage of evolution. Long term of independent evolution may have resulted in a huge difference in the accessory genome. Pronounced difference between the accessory genome of the two clades had been observed in a previous study (Howell et al., 2014).

Serovar 12 and 13 were found in relatively equal proportions in both clades in this study, but were identified predominantly in one clade in other literature (Howell et al., 2014). Most of the serovar 12 or 13 isolates in clade II were found harboring typical PVMs characteristics of the counterparts in clade I, which indicates that those isolates were probably originated from clade I and transformed into clade II isolates through homologous recombination at or around the site of iscR gene. Since iscR was the target gene for population structure analysis in this study, isolates that came across gene exchange at this site of the chromosome may be shifted from one clade to another. It is noteworthy that gene exchange among isolates within the same clade would be difficult to spot due to the similar genetic background. Higher rates of homologous recombination for the clinical isolates compared with the non-clinical isolates have been observed in H. parasuis (Vos & Didelot, 2008). More complex of PVMs profile were observed in isolates from multi-strain infected farm, indicating more frequent gene exchange in those isolates. Both higher prevalence and higher proportion of systemic isolates in serovar 12 and 13 increase the opportunity for them to contact with other isolates through co-infection, which may be part of the reason that leads to the higher rates of gene exchange in those serovars. Hence, extensive recombination within the species makes it harder to decipher the population structure of H. parasuis.

No isolate from healthy pigs was included for comparison in this study, and the absence of commensals was a possible reason for the low resolution of vtaA and ompP2 in differentiation of the isolates although these two genes were also found in all strains including both commensals and systemic isolates in other studies (Mullins et al., 2009; Howell et al., 2014). The common pathogens that co-infected with the H. parasuis isolates in our diagnostic center are porcine reproductive and respiratory syndrome virus, porcine circovirus type 2, Streptocccus suis, Bordetella bronchiseptica etc. Recent research showed that many previously suggested virulence factors were not appropriate markers of virulence (Howell et al., 2014; Turni, Singh & Blackall, 2018). In this study, a total of six systemic isolates harbored none of the PVMs, co-infection with other viral or bacterial pathogens were found for only three of the isolates. No correlation was found between the 10 selected PVMs and clinical virulence of the isolates. Those PVMs were probably serovar or clade specific genes emerged during evolution. To a certain extent, those PVMs may contribute to virulence, but they are not the genes that enable the organism to be virulent.

Of the 20 farms of multi-strain infection, systemic disease caused by multi-strains (ZJ10151-Su and ZJ10152-S2) was found in one farm, which stress the need for full assessment of the causative strains before treatment. Of the 86 lung isolates, a total of 24 isolates were isolated from lungs with presence of fibrinous inflammation, and the others were isolated from lungs with barely pneumonia. No difference was observed between the lung isolates from the two different clinical cases in respect of serovar distribution, PVMs profile and iscR phylogeny. Moreover, some of the pathogens such as S. suis and Mycoplasma hyorhinis could also cause lesions similar to Glässer’s disease, and we could not exclude the co-infection of all of these possible agents especially for M. hyorhinis in this study. Hence, virulence factors involved in invasion are still to be determined for the agent.

Conclusions

Iron-sulfur cluster regulator is a reliable marker for studying population structure of H. parasuis, while other factors should be included to avoid the interference of gene exchange. The two lineages of H. parasuis may have undergone independent evolution, but shows no difference in clinical virulence. Wide distribution of systemic isolates across the entire population poses new challenge for development of vaccine with better cross-protection. Our study provide new information for better deciphering the population structure of H. parasuis, which helps understanding the extreme diversity within this pathogenic bacterium.

Supplemental Information

Supplemental Information 1 Alignment of iscR protein sequences of 15 reference strains of H. parasuis using Lasergene (MegAlign).

IscR sequences of 15 reference strains originated from a previous study (28). Sequence name includes information on both name (e.g., Nagasaki) and serovar (e.g., “S5R” is short for serovar 5 reference strain) of the strain. IscR sequence of Nagasaki-S5R were used as markers, and residues that match Nagasaki-S5R exactly were hidden (as “ . ”). The clade distribution of each sequence type was indicated on the left.

Click here for additional data file.

Supplemental Information 2 Information of isolates used in this study.

Names, serovar, potential virulence markers, isolation sites of isolates are shown in this table.

Click here for additional data file.

We thank the members of the Swine Disease diagnostic Center of the institute of the animal husbandry and veterinary science for their underlying contributions.

Additional Information and Declarations

Competing Interests

Author Contributions

DNA Deposition

Data Availability

The authors declare that they have no competing interests.

Junxing Li conceived and designed the experiments, performed the experiments, contributed reagents/materials/analysis tools, prepared figures and/or tables, authored or reviewed drafts of the paper, approved the final draft.

Lihua Xu performed the experiments, prepared figures and/or tables.

Fei Su analyzed the data.

Bin Yu analyzed the data.

Xiufang Yuan conceived and designed the experiments, contributed reagents/materials/analysis tools, approved the final draft.

The following information was supplied regarding the deposition of DNA sequences:

The nucleotide sequences of iscR for the 148 isolates are available in GenBank. The accession numbers for the individual CDSs are MH982282–MH982429.

The following information was supplied regarding data availability:

The raw measurements are available in Dataset S1.

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
