# Peer review of "Association between iscR-based phylogeny, serovars and potential virulence markers of Haemophilus parasuis"

_PeerJ, doi:10.7717/peerj.6950_

## Round 0.1 · original submission · Major Revisions

As you can see, your submission raised a number of criticisms, and one reviewer even suggested that more wok is needed. Aside from the comments shown in the reviewer's reports, do not forget to look at the annotated manuscript.

If you are willing to make the work suggested by the reviewers and to correct your manuscript accordingly, I'll be pleased to see the revised version. When you submit it, do not forget to include a detailed and point-by-point rebuttal where you explain how and where the original submission has been modified, or justify why you did not follow all suggestions. Please, bear in mind that your revised version will undergo a new round of review (by the same or by different reviewers). I cannot, therefore, make any commitment about the final acceptance of your paper.

Reviewer 1 ·

Basic reporting

There is some speculation on horizontal gene transfer in the study. There is not an analysis, such as linkage disequilibrium or other, to measure or quantify it.

Some sentences are very difficult to interpret (Lines 189-190, lines 236-238))

The objective of section ‘Higher diversity of PVMs profiles were found in isolates from multi-strain infected farms’ is not clear. It seems to deal with strain diversity, but it is not clear by the wording used.

Discussion is too long and somewhat repetitive

Verb tense and plural/singulars forms need to be checked through the text.

Experimental design

One important aspect that the authors do not take into account when analyzing and discussing their data is the absence of non-virulent isolates in their collection. The collection is composed of only isolates from clinical samples. This is a major drawback of this study. Other studies that they use for comparison and discussion include non-clinical non-virulent isolates. This aspect need to be considered. In fact, the authors state in the introduction that H. parasuis can be commensal or pathogenic and it is important to differentiate between both status, but this is impossible if no commensal H. parasuis is included in the analysis. This absence of commensals could explain also the results with vtaA and OmpP2. This need to be taken into consideration in the discussion section, too. An extra information that could be included in the analysis is to investigate the clinical origin of lung isolates as coming from pneumonia cases or Glässer’s diseases cases. Sometimes lungs are used for diagnosis when the clinical picture is not barely a pneumonia, but systemic Glässer’s disease and this could be indicative of the invasive potential of the isolates.

Validity of the findings

Besides the comments about the isolate collection used in the study, some analysis on specific cases can be added to improve the discussion of the results:

Is the 16% of multi-strain infection associated with respiratory disease/isolation? This would be an interesting information, since it is considered that systemic disease is caused by a single strain.

It would be also very interesting to know if there are other pathogens involved in the clinical cases, especially in the cases when systemic isolates were found not to have putative virulence markers.

Additional comments

The title does not represent the content of the manuscript. The study shows similar results than previous studies on population structure of H. parasuis, and does not give new insights on clinical virulence.
This study reports the population structure obtained with IscR and the correlation of the clades with strain serovar and the presence of putative virulence markers. These data are interesting, but there are some important considerations that need to be addressed by the authors. (see above comments)

Section ‘Separation of clinical isolates into two deeply divergent groups by iscR-based phylogenetic analysis’: the results presented here in relation to the amino acid substitutions are difficult to follow since in the beginning (Fig 2A) the comparison are made against the sequence from Nagasaki strain, and therefore the interpretation is completely different to what is written. The comparison to other bacteria (Fig 2B), which defines the substitutions, should be presented first.

Reviewer 2 ·

Basic reporting

No comment

Experimental design

No comment

Validity of the findings

No comment

Additional comments

The manuscript Peer J 33889 describes an interesting study about the phylogeny and potential virulence markers of of Haemophilus parasuis, etiological agent of Glässer’s disease. Before being accepted some subjects must be revised:
• The English language should be improved (especially verbs) to ensure that an international audience can clearly understand the text. Some examples where the language could be improved include lines 30, 33, 34, 35, 72, 88, 92, 110, 129, 180, 187, 192, 211, 245, 260, 277, 292, 313, 314, 336, 349, 370, 381 among others

• Introduction section is too long and it must be shortened about one paragraph

• References in the text must be ordered chronologically, not alphabetically

• Line 101: “etc” cannot be used in this context. All sites from which samples were recovered must be cited

• Line 102: The author must indicate CO2 conditions of incubation

• Line 107: “Serovar” instead of “serova”

• Line 221: “clade” instead of “calde”

• Line 242: “respectively” must be deleted in this context

• Line 269: “recently” cannot be used when one of the references is dated on 2010

• Discussion section is too long and it must be shortened about one page

• Line 369: What is the meaning of serovars 4, 5 et al.?

• Line 378: “It shows that” must be deleted in this context

Annotated reviews are not available for download in order to protect the identity of reviewers who chose to remain anonymous.

---

## Round 0.2 · Minor Revisions

Your revised version has now been examined by the original reviewers. Both were partly but not fully satisfied with the revision since both pointed out that minor revision is still needed. Please, pay attention to their remarks (and look at the annotated manuscript) and submit a new version. I do hope that this new version will be final, but I cannot make any commitment at this stage. So, be careful to take each comment into account (and tell us what you did) or tell us why you did not.

Reviewer 1 ·

Basic reporting

Some improvement of the English would be appreciated

In some sections Vta gene is used, while the correct name of these genes is vtaA, and as for all gene names should be written in italics and lowercase. The same applies to other gene names.

Experimental design

The use of nasal cavity samples for diagnosis is not commonly used. Only samples from lesions should be considered clinical samples for Glässer’s disease. In fact figure 3 does not include this isolation site

Validity of the findings

The results are interesting and the conclusions and comparison to data from the literature are adequate

Additional comments

The manuscript has been improved

Reviewer 2 ·

Basic reporting

The authors have now corrected almost all the recommendations done in the first review of manuscript and, consequently, it should be accepted after these minor subjects are modified:
 Lines 53 and 54: References “Lawerence et al.” and “Brockmeier et al.” are not in an orthographic order
 Line 93: “And” must be added between “effusion” and “seroperitoneum”
 Line 175: “Data” (datum is singular, data is plural) instead of “datas”
 Line 182: “genes” instead of “gen”
 Line 256: “Our study was also consistent” instead of “Our study also consistent”

Experimental design

No comment

Validity of the findings

No comment

Additional comments

The authors have now corrected almost all the recommendations done in the first review of manuscript and, consequently, it should be accepted after these minor subjects are modified:
 Lines 53 and 54: References “Lawerence et al.” and “Brockmeier et al.” are not in an orthographic order
 Line 93: “And” must be added between “effusion” and “seroperitoneum”
 Line 175: “Data” (datum is singular, data is plural) instead of “datas”
 Line 182: “genes” instead of “gen”
 Line 256: “Our study was also consistent” instead of “Our study also consistent”

Annotated reviews are not available for download in order to protect the identity of reviewers who chose to remain anonymous.

---

## Round 0.3 · accepted · Accept

Thank you for making the final corrections. I believe your paper is now stronger and I wish you good success with this publication.